# Hand Assisted Laparoscopic Surgery for Colorectal Cancer: Surgical and Oncological Outcomes from a Single Tertiary Referral Centre

**DOI:** 10.3390/jcm11133781

**Published:** 2022-06-29

**Authors:** Narimantas Evaldas Samalavicius, Zygimantas Kuliesius, Robertas Stasys Samalavičius, Renatas Tikuisis, Edgaras Smolskas, Zilvinas Gricius, Povilas Kavaliauskas, Audrius Dulskas

**Affiliations:** 1Department of Surgery, Klaipeda University Hospital, 92288 Klaipeda, Lithuania; narimantas.samalavicius@gmail.com; 2Health Research and Innovation Science, Faculty of Health Sciences, Klaipeda University, 92294 Klaipeda, Lithuania; robertas.samalavicius@santa.lt; 3Institute of Clinical Medicine, Faculty of Medicine, Vilnius University, 03101 Vilnius, Lithuania; 4Department of Surgery, Republic Vilnius University Hospital, 28151 Vilnius, Lithuania; kzygimantas@yahoo.com (Z.K.); povilaskava@gmail.com (P.K.); 5Lithuania Emergency Medicine Center, Vilnius University Hospital Santaros Clinics, 08410 Vilnius, Lithuania; 6National Cancer Institute, 1 Santariskiu Str., 08406 Vilnius, Lithuania; renatas.tikuisis@nvi.lt (R.T.); zilvinas.gricius@gmail.com (Z.G.); 7Vilnius City Clinical Hospital, 57 Antakalnio Str., 10207 Vilnius, Lithuania; edgaras.smolskas@gmail.com

**Keywords:** colorectal cancer, hand-assisted surgery, laparoscopic surgery, complications, survival

## Abstract

The aim of this study was to report overall experience, perioperative and long-term survival results in a single tertiary referral center in Lithuania with hand assisted laparoscopic surgery (HALS) for colorectal cancer. A prospectively maintained database included 467 patients who underwent HALS for left-sided colon and rectal cancer, from April 2006 to October 2016. All those operations were performed by three consultant surgeons and nine surgical residents, in all cases assisted by one of the same consultant surgeons. There were 230 (49.25%) females, with an average age of 64 ± 9.7 years (range, 26–91 years). The procedures performed included 170 (36.4%) anterior rectal resections with partial mesorectal excision, 160 (34.26%) sigmoid colectomies, 81 (17.35%) left hemicolectomies, 45 (9.64%) low anterior rectal resections with total mesorectal excision, and 11 (2.25%) other procedures. Stage I colorectal cancer was found in 140 (29.98%) patients, 139 (29.76%) stage II, 152 (32.55%) stage III and 36 (7.71%) stage IV. There were five conversions to open surgery (1.1%). The mean postoperative hospital stay was 6.9 ± 3.4 days (range, 1–30 days). In total, 33 (7.06%) patients developed postoperative complications. The most common complications were small bowel obstruction (*n* = 6), anastomotic leakage (*n* = 5), intraabdominal abscess (*n* = 4) and dysuria (*n* = 4). There were two postoperative deaths (0.43%). Overall, 5-year survival for all TNM stages was 85.7%, 93.2% for stage I, 88.5% for stage II and 76.3% for stage III. Hand assisted colorectal surgery for left-sided colon and rectal cancer in a single tertiary referral center was feasible and safe, having all the advantages of minimally invasive surgery, with good perioperative parameters, adequate oncological quality and excellent survival.

## 1. Introduction

Laparoscopic colorectal resections were first reported in 1991 [1], very soon after laparoscopic surgery was introduced into surgical practice. Quite shortly after this, a possibility of performing laparoscopic colorectal resections while using a mini-laparotomy for hand insertion was described [2]. Authors noted, that in comparison with open resection, this type of surgery was related with less pain, better cosmetic results, earlier food intake and shorter hospital stay, though the operative time in the hand assisted group was longer. In other words, it seemed like hand assisted surgery may give similar advantages as straight laparoscopic surgery in comparison with open surgery.

In the light of that, the first had insertion devices were introduced [3,4,5]. This enabled rapid progress and spread of hand assisted laparoscopic surgery, including in the colorectal field. At our institution, hand assisted laparoscopic surgery was introduced for left-sided colon and rectal cancer in 2006, and our initial experience with this technique has been already previously published [6,7,8].

We report our overall experience in a single tertiary referral center in Lithuania with laparoscopic hand assisted surgery for colorectal cancer.

## 2. Materials and Methods

### 2.1. Patients

The institutional review board approved the study.

We conducted a retrospective study of prospectively-collected data in a single tertiary care institution. A prospectively maintained database contains all patients who underwent HALS for left-sided colon and rectal cancer at the Department of Surgical Oncology, National Cancer Institute, from April 2006 to October 2016. All consenting patients aged 18 years or older with histologically-confirmed invasive cancer of the descending colon, the sigmoid colon and the upper and the middle rectum were included in this study. Data on the following clinical factors were collected: gender, age, comorbidities, prior abdominal surgery, cancer stage by TNM staging system, the operation performed, operative time, conversion, number of lymph nodes harvested, length of removed specimen, distance from the tumor to the distal and proximal ends of removed specimen, postoperative complications, length of hospital stay, time of first hospitalization, time of patient’s last visit or death and time of disease progression.

Length of hospital stay was defined as the number of nights the patient spent from the day of surgery until discharge. Anastomotic leak was defined as any clinical or radiological evidence of dehiscence of the anastomosis: the presence of peritonitis caused by anastomosis dehiscence, the presence of feculent discharge from the drainage tube or the presence of abscess with demonstration of leak. These were also confirmed by radiography from drainage tube, hydro soluble enema or CT-guided abscess drainage except the cases with obvious feculent discharge from the drainage tube [9]. A mass in the pelvis around or in anastomosis site found by clinical, endoscopic, radiologic, pathologic examination or autopsy was defined as local recurrence. Similarly, distant recurrence was defined as tumor growth in any lymph node outside the abdomen, or in any other organ.

### 2.2. Surgical Technique

All laparoscopic procedures were performed by skilled surgeon or a trainee assisted by a skilled surgeon in a clear and same for every case standardized manner [2,3]. Under general anesthesia with the patient in a supine horizontal position with legs outstretched, the body fixed to the operating table and the surgeon standing on the patient right, a 6-to 6.5-cm-long trans-umbilical incision was performed for the Dextrus Endopath (Ethicon Endo-Surgery LLC, Guaynabo, PR, USA) hand-port device insertion. The HALS resection was accomplished with this hand port and two or three (third one for the splenic flexure mobilization and/or dissection in rectal cancer surgery) additional ports. Under hand control, a 12-mm trocar was inserted 2–3 cm towards the midline and 2–3 cm below the right anterior superior iliac spine, and a 10-mm trocar was inserted at the level of the right midclavicular line a few centimeters above the umbilicus (with a camera port to allow the visualization of both the splenic flexure and transverse colon and the pelvic area); in addition, when needed, a 5-mm trocar was inserted into the left lateral quadrant a few centimeters above and towards the midline from the anterior superior iliac spine. Mobilization began with moving of the descending colon upwards to splenic flexure and left side of the transverse colon using a hand and a harmonic scalpel (cranial part elevated and patient turned to the right). During this part, the operator was standing between outstretched legs. When this part was finalized, the operator moved to the right side of the patient (same as assistant surgeon) and continued mobilization of the sigmoid colon, then lifting the rectosigmoid colon to the level of the promontory with the superior rectal vessels and mobilizing from the left side using a 12-mm trocar for the harmonic scalpel and visualizing the left ureter (caudal part elevated and turned to the right). Then the inferior mesenteric artery was mobilized and ligated using titanium 10-mm clips at 1–2 cm from the aorta, and the inferior mesenteric vein was mobilized and ligated with same clips at the level of the ligament of Treitz. The specimen was divided at the level of the promontory (for left-sided cancer), or 5 cm below the cancer in anterior rectal resection with partial mesorectal excision (PME), or 2 cm in anterior rectal resection with total mesorectal excision (TME), using a 60 mm endoscopic linear stapler (one cartridge was generally enough, and rarely two cartridges were necessary) and removed through the hand-port incision, and an anastomosis was fashioned laparoscopically using a double-stapling technique. The water–air leak test was performed, and the rings from the stapler were examined to ensure completeness. A drain was not routinely used (more often in rectal cancer surgery). The fascia was closed at the level of the 12-mm trocar with a single interrupted suture, and the hand port with a running polydioxanone 0 suture. The skin incisions were closed with interrupted sutures.

### 2.3. Statistics

Statistical analyses were performed using IBM SPSS Statistics for Windows, Version 23.0 (IBM Corp., Armonk, NY, USA). All data are presented as mean ± standard deviation for parametric, and median for nonparametric data. Survival curves were produced by means of the Kaplan–Meier model. The log-rank test was used to evaluate the statistical differences between the survival curves.

## 3. Results

Over a period of 10 years, from 2006 to 2016, 467 HALS colorectal resections were performed. All the operations were performed by three consultant surgeons and nine surgical residents, in all cases assisted by one of the same consultant surgeons. There were 230 (49.25%) females and 237 (50.75%) males, with an average age of 64 ± 9.7 years (range, 26–91 years) (Table 1). A total of 228 patients (48.82%) had comorbidities: 187 of them were cardiovascular, 26 had diabetes, 16 were pulmonary and 9 were renal. Thirty-five had other various comorbidities such as psoriasis, hypothyroidism, Parkinson’s disease and other. No statistically significant comorbidity prevalence was observed among different gender. One hundred and nine (23.34%) patients had experienced a prior abdominal surgery.

We have performed these HALS procedures: 170 (36.4%) anterior rectal resection with partial mesorectal excision, 160 (34.26%) sigmoid colectomy, 81 (17.35%) left hemicolectomy, 45 (9.64%) anterior rectal resection with total mesorectal excision and 11 (2.35%) other procedures. There were five conversions to open surgery (1.1%), all due to technical difficulties.

Clinical data of intraoperative outcomes and cancer stage by TNM staging system is shown in Table 2. The mean postoperative hospital stay was 6.9 ± 3.4 days (range, 1–30 days). In total, 33 (7.06%) patients developed postoperative complications; four of them developed two different complications each. The most common complications were small bowel obstruction (6 patients), anastomotic leakage (*n* = 5), intraabdominal abscess (*n* = 4) and dysuria (*n* = 4). There were two postoperative deaths (0.43%): one patient due to pulmonary embolism, another due to septic pneumonia.

Overall 5-year survival in 467 patients was 97.7% (Figure 1), whilst 5-year OS was 85.7% (Figure 2). The 1-year survival rates according to TNM stage were: stage I—99.2%, stage II—96.4%, stage III—97.4%; there was no significant difference between groups (*p* = 0.321) (Figure 3). The 5-year survival rates according to TNM stage were significantly different: stage I—93.2%, stage II—88.5%, stage III—76.3% (*p* = 0.001) (Figure 4). DFS for 1 year were: stage I—96.7%, stage II—95.1%, stage III—94.1% (*p* = 0.67). DFS for 5 years were: stage I—88.8%, stage II—86.4%, stage III—85.7%, (*p* = 0.85). There were no significant differences between groups. For the 17 patients, progression of the disease was diagnosed more than 5 years later (from 5.9 to 9.45 years). The 1-year survival rate for patients with rectal cancer was 97.5% and for all other patients it was 97.8% (*p* = 0.842). The 5-years survival rate for rectal cancer patients and all other patients were 84.5% and 86.6% respectively (*p* = 0.432).

## 4. Discussion

In this paper, we have not compared HALS surgery with either open or straight laparoscopic colorectal cancer surgery, performed in our institution. In our experience HALS to some surgeons was a bridge to straight laparoscopic surgery from open surgery, even though for some operations (like right hemicolectomy, or abdominoperineal resection) we never implemented HALS technique as for these operations straight laparoscopy was used from the very beginning.

Comparison of open and HALS surgery was of importance trying to delineate the place of HALS in the era of minimally invasive surgery. Kang et al. [10] compared open surgery and HALS and found that HALS produces better therapeutic results in terms of perioperative parameters. Orenstein et al. [11] concluded that HALS is advantageous compared to open surgery in an elective setting. A similar conclusion was reached in the first systematic review on this subject [12], in addition stating that HALS is an excellent treatment option when specimen extraction is needed, and allowed to avoid some of the disadvantages of straight laparoscopic surgery, having shorter operative time, less conversions. Looking specifically to HALS and open surgery results for rectal cancer, Liu et al. [13] observed the former to be superior in all perioperative parameters, and equal oncological results, though follow up time was quite short in their study. An important conclusion came from analyzing 2012 colectomy-targeted American College of Surgeons National Surgical Quality Improvement Program database, matching 1740 open colonic resections with 1740 HALS [14], concluding that HALS was related with shorter hospital stay, less reoperations, less readmissions, less ileus and less overall complication rate; it could be a bridge to straight laparoscopy and a tool in difficult cases. Another similar study queried the 2012–2013 National Surgery Quality Improvement Program for adults undergoing elective HALS or open colectomy. After propensity matching, short-term outcomes were compared between 2707 open and 6084 HALS. Improved perioperative outcomes in HALS group were observed with no increase in operative time [15]. In a relatively recent study from Japan [16], the authors presented their experience with HALS surgery for stage I-III colorectal cancer matching with a similar group where open surgery was performed, and found HALS to be performed more safely and to give superior immediate results and same oncologic outcome; they thought this type of surgery may be of use in small and medium-sized Japanese hospitals.

Obviously, the most interesting is the comparison of HALS and straight laparoscopic surgery. The first RCT came out in 2002 [17] comparing 27 cases in each group. It was noted that HALS helps in a difficult situation, less likely to need conversion, maintains features on oncological and laparoscopic surgery for the similar cost. In a comparative study, Ringley et al. [18] observed shorter operative time and better lymph node harvest, and only 1 cm longer overall incision length in HALS group. In a multicenter randomized trial, Marcelo et al. [19] demonstrated shorter operative times in left-sided and total abdominal colectomies in favor of HALS, and similar clinical outcomes. In a study from the Mayo clinic comparing straight and HALS colorectal resections during one calendar year, it was noted that HALS was used for more complex procedures while maintaining patient short-term benefits [20]. Interestingly, from the Mayo clinic a largest single center series on HALS were reported (a total of 1103) [21]. Our series are the second biggest reported in the literature according to the number of patients who underwent HALS colorectal resections. The authors of a more recent report included 1307 patients from nine studies that were found eligible for meta-analysis and concluded that both types of surgery had similar intraoperative, postoperative and survival outcomes except longer incision in HALS group [22]. In an experimental study using ProMIS simulator it was nicely demonstrated that reduction of the operative time did not cause more intraoperative errors in HALS surgery, but did in straight laparoscopy [23].

When it comes to training, there is one judgement in the literature stating that approximately 100 hand-assisted surgeries are needed to be performed by a single surgeon to achieve technical proficiency [24]. Our experience does not contradict this statement. However, we do very much agree with Ozturk et al. who stated that though operative time decreased with experience, there was learning curve for the quality of HALS surgery [25].

Another important aspect of HALS is operative costs, since a special disposable hand insertion device is required. Two independent studies from different countries found certain differences compared with straight laparoscopic surgery regarding consumables, but overall operation cost was not different [26,27].

It should be brought to attention that in this unselected patient population excellent 5-year overall survival and disease-free survival results were achieved, both as such, and especially comparing to Lithuanian result in general. There are very few studies regarding survival of colorectal cancer patients in Lithuania related to the period of our study [28,29,30]. A study from three Lithuanian cancer care institutions compared perioperative and long-term survival results in Lithuania in 2005 and 2010. An improvement was noted in overall survival comparing the two periods: for stage I, 82.8% and 83%, stage II, 76% and 78.9% and stage III, 49.2% and 56.4%; overall survival of colorectal cancer patients for all stages was 52.1% and 63.1% in this study [28]. In a population-based study, overall survival of colorectal cancer patients in Lithuania was 37.9% for the period 1998–2002 and increased during the period of 2008–2012 to 51.5%; in the later period patients with localized disease (T1-T4) had a survival rate of 78.6%. In our patient population overall 5-year survival for all stages was 85.6%; for localized disease stage I, 93.2%, and stage II, 88.5%; even for lymph node positive patients, overall 5-years survival was 76.3%. In our opinion, the major impact was standardization of surgical technique from the very first patient regardless for the operating surgeon.

Obviously, our study is limited by a retrospective single center experience with a lacking control group (open and/or straight laparoscopy). However, data from a tertiary center with long-term results adds significance to the scientific body.

## 5. Conclusions

Hand assisted colorectal surgery for left-sided colon and rectal cancer in a single tertiary referral center was feasible and safe, having all advantages of minimally invasive surgery, with good perioperative parameters, adequate oncological quality and excellent survival.

## Figures and Tables

**Figure 1 jcm-11-03781-f001:**
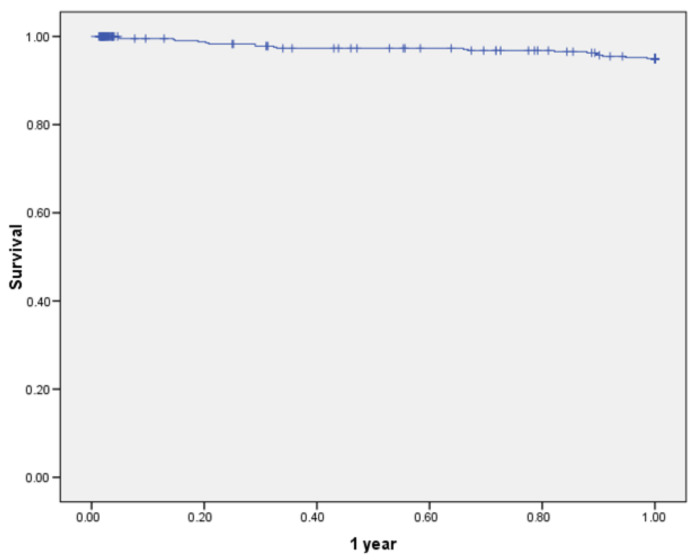
Overall 1-year survival of patients undergoing hand-assisted laparoscopic surgery for cancer (all stages).

**Figure 2 jcm-11-03781-f002:**
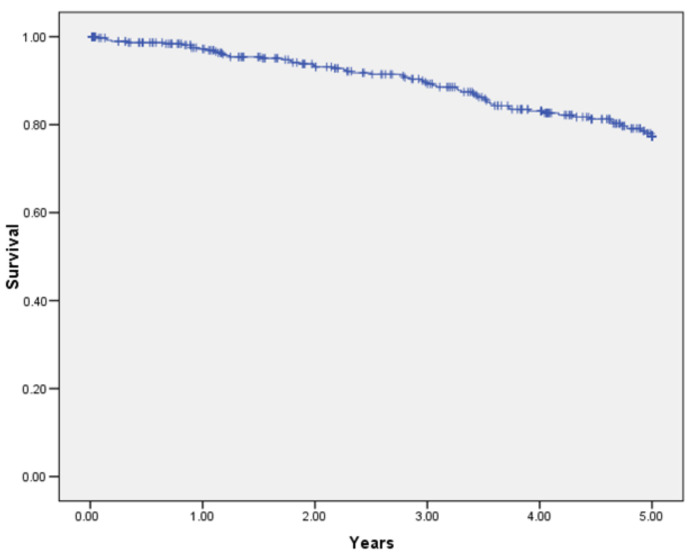
Overall 5-year survival of patients undergoing hand-assisted laparoscopic surgery for cancer (all stages).

**Figure 3 jcm-11-03781-f003:**
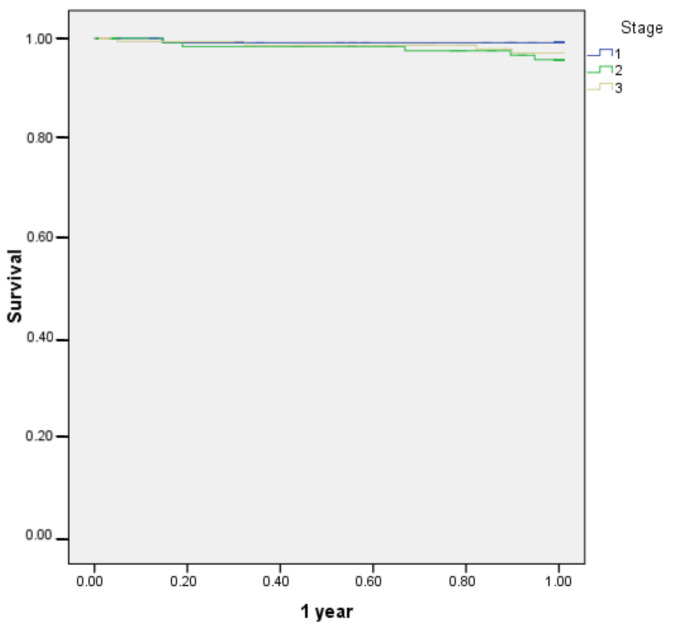
1-year survival according to TNM stage of patients undergoing hand-assisted laparoscopic surgery for cancer (stage I–III).

**Figure 4 jcm-11-03781-f004:**
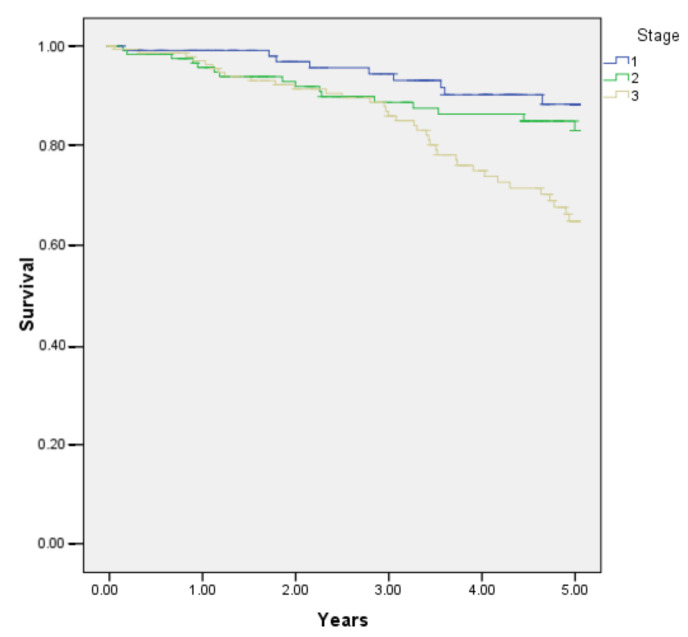
5-year survival according to TNM stage of patients undergoing hand-assisted laparoscopic surgery for cancer (stage I–III).

**Table 1 jcm-11-03781-t001:** Demographic data of included patients.

Parameters	Results
Patients, *n*	467
Gender, *n* (%):	
Male	237 (50.75)
Female	230 (49.25)
Age, mean ± SD, range (years)	64 ± 9.7 (26–91)
Comorbidities, *n*:	
Cardiovascular	187
Diabetes	26
Pulmonary	16
Renal	9
Other	35
Prior abdominal surgery, *n* (%)	109 (23.34)

**Table 2 jcm-11-03781-t002:** Intraoperative outcomes and cancer stage by TNM staging system.

Parameter	Results
Operative time, mean ± SD, range (min)	112 ± 44 (30–320)
Length of specimen, mean ± SD (cm)	18.6 ± 8.4
Distance from tumor to distal end of specimen, mean ± SD (cm)	5.2 ± 5.5
Distance from tumor to proximal end of specimen, mean ± SD (cm)	10.3 ± 5.3
No. of harvested lymph nodes, *n*	17 ± 12
TNM stage, *n* (%)	
I	140 (29.98)
II	139 (29.76)
III	152 (32.55)
IV	36 (7.71)

## Data Availability

The data presented in this study are available on request from the corresponding author. The data are not publicly available due to ethical reasons.

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
