# Peer review of "Hand Assisted Laparoscopic Surgery for Colorectal Cancer: Surgical and Oncological Outcomes from a Single Tertiary Referral Centre"

_jcm, 2022, doi:10.3390/jcm11133781_

Round 1
Reviewer 1 Report
This paper is a very interesting discussion of HALS, which falls somewhere between conventional open laparotomy and pure-laparoscopic surgery. And I consider it potentially acceptable for publication.
1) It would be more important to consider only the rectum. This is because the rectal cancer surgery, unlike the colon, is more difficult to treat and has a 5-10% lower survival rate with pelvic local recurrence.
2) I think it would be better to extract open laparotomy as a comparison. If the authors consider only HALS without background bias, I think that at least the 5-year relapse-free survival rate and 5-year overall survival rate at stages 2 and 3 excluding stages 1 and 4 should be evaluated.
3) I agree that HALS is safe, reliable, and minimally invasive, but I think you should definitely try to expand it to right colonic surgery including right hemicolectomy.
Author Response
A point-by-point response to the editor's comments:
Dear Editor,
Thank you for your letter and constructive comments concerning our manuscript entitled “Hand assisted laparoscopic surgery for colorectal cancer: surgical and oncological outcomes from a single tertiary referral centre”. The paper was revised substantially. Following changes have been made. They are as follows:
A point-by-point response to the editor's comments:
Dear Editor,
Thank you for your letter and constructive comments concerning our manuscript entitled “Hand assisted laparoscopic surgery for colorectal cancer: surgical and oncological outcomes from a single tertiary referral centre”. The paper was revised substantially. Following changes have been made. They are as follows:
This paper is a very interesting discussion of HALS, which falls somewhere between conventional open laparotomy and pure-laparoscopic surgery. And I consider it potentially acceptable for publication.
1) It would be more important to consider only the rectum. This is because the rectal cancer surgery, unlike the colon, is more difficult to treat and has a 5-10% lower survival rate with pelvic local recurrence.
Totally agree. We separated the survivals of colon and rectum cancers (we have included this in the Results part).
2) I think it would be better to extract open laparotomy as a comparison. If the authors consider only HALS without background bias, I think that at least the 5-year relapse-free survival rate and 5-year overall survival rate at stages 2 and 3 excluding stages 1 and 4 should be evaluated.
We thought of comparing HALS to open surgery, but our goal was to see our experience with HALS only. Probably then we should compare with straight laparoscopy as well. Finally, the survival analysis included separate curves for stage I, II and III diseases.
3) I agree that HALS is safe, reliable, and minimally invasive, but I think you should definitely try to expand it to right colonic surgery including right hemicolectomy.
We agree that HALS can be done for right hemicolectomy also. However, as most of our team surgeons are right-handed, for them this procedure was a great struggle.
Reviewer 2 Report
1. Each comorbidity was equally presented between males and females? Have you noticed a preferential presence of any comorbidity in males and females?
2. Percentages should be checked again
3. Discussion: Lanes 206, 207: "Our series are second biggest reported in the literature according to the number of patients who underwent HALS colorectal resections",
However at the end of discussion it is written: Lanes 240, 241: "our study is limited by a small sample size and single center experience 240 lacking control group"
Since it is the second reported series, why it is considered as a small sample size? Could you explain?

Author Response
A point-by-point response to the editor's comments:
Dear Editor,
Thank you for your letter and constructive comments concerning our manuscript entitled “Hand assisted laparoscopic surgery for colorectal cancer: surgical and oncological outcomes from a single tertiary referral centre”. The paper was revised substantially. Following changes have been made. They are as follows:
- Each comorbidity was equally presented between males and females? Have you noticed a preferential presence of any comorbidity in males and females?
No, we did not find any statistically significant difference between the males and females (have included in the Results section).
- Percentages should be checked again
Percentages were corrected (marked in the text). Thank you for the notification.
- Discussion: Lanes 206, 207: "Our series are second biggest reported in the literature according to the number of patients who underwent HALS colorectal resections". However at the end of discussion it is written: Lanes 240, 241: "our study is limited by a small sample size and single center experience 240 lacking control group". Since it is the second reported series, why it is considered as a small sample size? Could you explain?
Thank you for a great comment. We have deleted the limitation “small sample size” (marked in the text).
Thank you very much. The manuscript improved a lot.
Sincerely
Audrius Dulskas